# Sulconazole-Loaded Solid Lipid Nanoparticles for Enhanced Antifungal Activity: In Vitro and In Vivo Approach

**DOI:** 10.3390/molecules28227508

**Published:** 2023-11-09

**Authors:** Ayesha Samee, Faisal Usman, Tanveer A. Wani, Mudassir Farooq, Hamid Saeed Shah, Ibrahim Javed, Hassan Ahmad, Riffat Khan, Seema Zargar, Safina Kausar

**Affiliations:** 1Department of Pharmaceutics, Faculty of Pharmacy, Bahauddin Zakariya University, Multan 66000, Pakistan; aashifarhat@live.com (A.S.); safinakausar555@gmail.com (S.K.); 2Department of Pharmaceutical Chemistry, College of Pharmacy, King Saud University, P.O. Box 2457, Riyadh 11451, Saudi Arabia; twani@ksu.edu.sa; 3Department of Manufacturing Pharmacy, Faculty of Pharmacy, Mahidol University, Bangkok 10400, Thailand; mudassir.far@student.mahidol.ac.th; 4Institute of Pharmaceutical Sciences, University of Veterinary and Animal Sciences, Syed Abdul Qadir Jillani (Out Fall) Road, Lahore 54000, Pakistan; 5Center for Pharmaceutical Innovation, Clinical and Health Sciences, The University of South Australia, North Terrace, Adelaide 5000, Australia; ibrahim.javed@unisa.edu.au; 6Faculty of Pharmaceutical Sciences, University of Central Punjab, 1-Khayaban.e. Jinnah Road, Johar Town, Lahore 54000, Pakistan; h.ahmad@ucp.edu.pk; 7College of Pharmacy, University of Sargodha, Sargodha 40100, Pakistan; khanriffat392@gmail.com; 8Department of Biochemistry, College of Sciences, King Saud University, P.O. Box 22452, Riyadh 11451, Saudi Arabia; szargar@ksu.edu.sa

**Keywords:** sulconazole, solid lipid nanoparticles, anti-fungal gel, histopathology

## Abstract

Solid lipid nanoparticles (SLNs) have the advantages of a cell-specific delivery and sustained release of hydrophobic drugs that can be exploited against infectious diseases. The topical delivery of hydrophobic drugs needs pharmaceutical strategies to enhance drug permeation, which is a challenge faced by conventional formulations containing a drug suspended in gel, creams or ointments. We report the fabrication and optimization of SLNs with sulconazole (SCZ) as a model hydrophobic drug and then a formulation of an SLN-based topical gel against fungal infections. The SLNs were optimized through excipients of glyceryl monostearate and Phospholipon^®^ 90 H as lipids and tween 20 as a surfactant for its size, drug entrapment and sustained release and resistance against aggregation. The SCZ-SLNs were physically characterized for their particle size (89.81 ± 2.64), polydispersity index (0.311 ± 0.07), zeta potential (−26.98 ± 1.19) and encapsulation efficiency (86.52 ± 0.53). The SCZ-SLNs showed sustained release of 85.29% drug at the 12 h timepoint. The TEM results demonstrated spherical morphology, while DSC, XRD and FTIR showed the compatibility of the drug inside SLNs. SCZ-SLNs were incorporated into a gel using carbopol and were further optimized for their rheological behavior, pH, homogeneity and spreadability on the skin. The antifungal activity against *Candida albicans* and *Trichophyton rubrum* was increased in comparison to a SCZ carbopol-based gel. In vivo antifungal activity in rabbits presented faster healing of skin fungal infections. The histopathological examination of the treated skin from rabbits presented restoration of the dermal architecture. In summary, the approach of formulating SLNs into a topical gel presented an advantageous drug delivery system against mycosis.

## 1. Introduction

Sulconazole (SCZ) is an imidazole derivative belonging to Biopharmaceutical Classification System (BCS) Class II with broad-spectrum antifungal activity. SCZ is a lipophilic drug (logP, 6.06) with poor water solubility (1.29 µg/mL) and a molecular weight of 397.755 g/mol and a pKa value of 6.78 (strong base) [1]. SCZ has fungistatic, fungicidal, antiparasitic, antibacterial and anticancer properties. It interferes with synthesizing ergosterol, which is the essential bioregulator of fungal cell membrane integrity and fluidity [2]. The antiparasitic activity of SCZ is believed to act through inhibition of CYP-450s, which are essential for protozoan and helminth parasites [3]. The antibacterial activity of SCZ may be due to the inhibition of bacterial (flavoHb) flavor hemoglobin. It also inhibits DNA gyrase and topoisomerase IV [4].

Globally, the most common superficial and deep-cutaneous fungal infections are dermatophytosis, pityriasis versicolor and candidiasis. Other cutaneous fungal infections include blastomycosis, mycetoma, chronoblastomycosis, mucormycosis and sporotrichosis. Dermatophytosis possesses a worldwide distribution and constitutes a serious health threat [5]. Social, geographic and environmental variables influence its occurrence, and its epidemiological features can alter over time [6]. People migrating more frequently, living in substandard conditions and using corticosteroids, antibiotics and frequent antineoplastic medications are factors that contribute to its occurrence [7].

Topical antifungal formulations are often preferred over systemic antifungal agents for treating fungal skin infections because they allow localized drug delivery to the affected sites, decreasing adverse effects and increasing patient compliance [8]. Because the fungal pathogens dwell on or in the stratum corneum (Sc), the antifungal medicine must be concentrated adequately to limit the fungal infection. SCZ topical formulations currently available in the market can be administered twice daily. The available formulation cannot prolong the drug’s residence time over the skin, necessitating a longer therapy term. Dermatophytes infiltrate the keratinized tissue of the Sc, causing cutaneous seborrheic dermatitis with significant psychological, social and economic repercussions for sufferers [9]. Most fungal infections are found in the lower epidermal (stratum basale layer), superficial area and vellus hairs as fungal cells have a specific efflux mechanism that results in drug resistance [10]. Suboptimal concentration at the site of infection, poor adherence to medication, ineffective delivery methods and an active efflux mechanism result in recurrent occurrences, chronic infections and the development of significant drug resistance. Due to long-term therapy and patient noncompliance with the typical dose form, dermatophyte-mediated intracellular infections are crucial to treat [11]. These sites of infection are inaccessible to typical formulations, resulting in low treatment effectiveness. The inability of topical antifungal medicines to enter the skin efficiently is the greatest barrier to their efficacy. To effectively treat cutaneous fungal infections, the therapeutic substance must reach the site of infection through the Sc sufficiently to elicit the pharmacological response [12].

Novel drug delivery systems, including liposomes, nanoemulsions and lipid nanoparticles, are promising carriers to develop topical antifungal formulations to obtain the targeted delivery of drugs and maintain the effective therapeutic concentration of drugs at the target site [13]. Solid lipid nanoparticles (SLNs) are composed of physiologically tolerant, able, biodegradable and biocompatible surfactants [14] for topical controlled and sustained delivery of encapsulated antifungal drugs to the skin. Since SLNs have a strong affinity for Sc, they facilitate drug permeation and localization, particularly for lipophilic drugs [15]. Their 10–1000 nm particle size indicates a large surface area, prolonged drug release and quick cell uptake [16,17]. SLNs have the ability to create an occlusive, invisible film on the skin that prevents water loss while assuring continuous drug release over an extended period of time. However, the low viscosity of the SLN’s dispersion can be managed by incorporating the SLNs into the gel to obtain a semisolid consistency of the final product. The cross-networking of the gel prevents the polymorphic transition and aggregation of SLNs. The composition of SLNs includes Phospholipon^®^ 90 H, glyceryl monostearate (GMS) and tween 20. Phospholipon^®^ 90 H is a biocompatible phospholipid that can help stabilize SLNs by developing a protective layer around the lipid nanoparticles, preventing aggregation and maintaining their stability. It can improve the encapsulation and delivery of lipophilic drugs due to their lipophilic nature and ability to solubilize them within the lipid matrix of SLNs [18]. GMS is a lipid-based biodegradable compound that helps to solidify the lipid core of SLNs, providing structural stability to the nanoparticles. It enables the sustained release of the entrapped drug by retarding its diffusion from the lipid matrix, thereby improving the therapeutic efficacy, and also acts as a surfactant [19]. Tween 20 is a non-ionic surfactant that assists in emulsification during SLN preparation. It reduces interfacial tension and enables the formation of smaller, more uniform nanoparticles. This study aimed to design and evaluate SCZ-loaded SLNs (SCZ-SLNs) to treat fungal skin infections better.

## 2. Results

The SCZ-SLN formulations (SLN1-SLN5) were prepared, characterized and evaluated.

### 2.1. Characterization of SCZ-SLNs

#### 2.1.1. Particle Size, Polydispersity Index and Zeta Potential

The particle size of the formulations was between 60.31 and 194.9 nm (Table 1). The PDI of all formulations was less than 0.5, indicating mono-dispersity of a stable vesicular system with less propensity for particle aggregation. The Zeta potential values of the formulations were between −16.40 and −28.52 mV. The particle size of the optimized formulation SLN2 is shown in Table 1.

#### 2.1.2. FTIR Spectroscopy

An FTIR analysis indicated the compatibility of SCZ, Phospholipon^®^ 90 H and GMS inside an SLN2 formulation. Spectra of SCZ (Figure 1, spectra A) showed the peaks of the out-of-plane bending of phenyl and imidazole deformation at 631 cm^−1^, and the peaks at 768.99 cm^−1^ indicated the out-of-plane bending of C-H aromatic and hetero-aromatic rings. In contrast, in-plane bending of imidazole ring deformation and C-H aromatic rings were demonstrated by the peaks at 859 cm^−1^ and 1089 cm^−1^, respectively. The peaks at 1335 cm^−1^ and 1382 cm^−1^ correspond to the stretching of the C-N imidazole ring. In Phospholipon^®^ 90 H spectra (Figure 1, spectra B), the peak at 1070 cm^−1^ reveals the stretching vibration of the C-O bond in the phosphate group (-PO-O-). It confirms the presence of the phosphate group in Phospholipon^®^ 90 H and, in addition, a C=O stretching band at 1732 cm^−1^ in the fatty acid ester and a P=O stretching band at 1236 cm^−1^. The peak at 2919 cm^−1^ gives information about the stretching vibration of the C-H bonds in the aliphatic chains. It indicates the presence of the fatty acid chains in Phospholipon^®^ 90 H. The GMS spectra (Figure 1, spectra C) illustrates a sharp peak at 1049 cm^−1^, suggesting stretching vibration of the C-O bond in the ester group (-COO-). The peak at 1728 cm^−1^ shows how the ester group’s C=O bond vibrates in a stretching manner, while the small peak at 2361 cm^−1^ and sharp peak at 2919 cm^−1^ indicate the presence of a methyl and methylene group in the stearic part of GMS. In the fabricated formulation (Figure 1, spectra D), the observed peaks at 1090 cm^−1^, 1450 cm^−1^, 1725 cm^−1^ and 2918 cm^−1^ indicate the presence of ester linkages, aromatic compounds, alkyne groups and fatty acid chains.

#### 2.1.3. Transmission Electron Microscopy (TEM)

The SCZ-SLNs’ TEM images show that the particles had globular, uniform, and smooth morphologies. The mean diameters of SCZ-SLNs ranged from 60 nm to 200 nm, as illustrated in Figure 2.

#### 2.1.4. Thermal Analysis

The DSC of SCZ exhibits a sharp exothermic peak at 178 °C (Figure 3A), suggesting a transformation such as crystallization or decomposition. The small endothermic peak at 290 °C indicates a phase transition, possibly from solid to a liquid or gaseous state. Figure 3B illustrates that pure Phospholipon^®^ 90 H showed two endothermic peaks at 126 °C and 410 °C, respectively. The first peak is related to the melting of the phospholipid molecules, and the later peak indicates a controlled phase transition from a gel-like phase to a liquid crystalline structure through crystalline or isomeric changes along the carbon chain [20]. A DSC thermogram of GMS (the primary ingredient of SLNs) showed an abrupt endothermic peak at 62.5 °C (Figure 3C), indicating melting endotherms. Similar findings were reported by A. R. Gardouh on the DSC of GMS, showing a melting endotherm of 60 °C [21]. The optimized SLN 2 formulation exhibits a distinct endothermic peak at 67 °C (Figure 3D), indicating a phase transition that is potentially related to the interaction between the components. An exothermic peak at 164 °C also suggests a transformation such as crystallization or decomposition. Another exothermic peak is observed at a higher temperature of 423 °C, which could be attributed to further decomposition or structural changes in the mixture. 

The TGA graph shows that SCZ undergoes partial dehydration associated with the compound’s structure at a temperature up to 178.32 °C. Within the 178 °C to 434.50 °C range, the significant weight loss of 63.61% suggests more extensive decomposition or degradation. Phospholipon^®^ 90 H displayed a weight loss of 16.56% at 316.71 °C and 71.04% at 466.93 °C. It could involve various processes such as the thermal decomposition of the Phospholipon^®^ 90 H fragmentation of the molecule, formation of volatile by-products or release of gases. GMS exhibited a weight loss of 12.47% at 289.13 °C and 80.53% at 445.46 °C. The SLN2 illustrated a weight loss of 12.67% at 264.13 °C and 68.52% at 391.46 °C, as shown in Figure 4. Overall, significant weight loss indicates the exothermic nature of the ingredients and formulation.

#### 2.1.5. X-ray Diffraction

X-ray diffraction is a non-destructive and very versatile method to characterize the crystalline and amorphous phases of the material. An XRD of SCZ, glyceryl monostearate (GMS), Phospholipon^®^ 90 H and the freeze-dried SLN2 of the optimized formulation was carried out (Figure 5).

The optimized formulation showed a sharp peak of intensity of 951 a.u. at 2θ = 24°, suggesting well-defined crystal planes. The peak positions differ from the components, indicating probable interactions or changes in the crystalline structure following mixing. GMS presented peaks of intensity of 4960 a.u. and 5540 a.u., at angles 20° and 25°, respectively, suggesting a well-defined crystal. Sharpness denotes strong crystallinity. Phospholipon^®^ 90 H’s sharp peak of intensity of 3806 a.u. at 2θ = 22° indicates its high crystallinity. SCZ illustrated peaks of intensity at angles 13° and 26° of 225 a.u. and 412 a.u., respectively. Sharp peaks imply well-defined crystal planes in combination.

#### 2.1.6. Encapsulation Efficiency (EE) and % Production Yield

The EE (%) values were in the range of 77.16–86.52% due to the lipophilic drug’s high affinity for the lipidic material. The EE (%) for the optimized SLN2 formulation was 86.52%. The % yields of production obtained were relatively high and were in the range of 80.16–91.35%, as presented in Table 2.

### 2.2. Characterization of SCZ-SLNs Gel

#### 2.2.1. pH, Homogeneity and Grittiness

All fabricated SCZ-SLNs gel formulations were white, homogeneous, were of a semisolid consistency and they could all be dispersed over the skin’s surface. All fabricated SCZ-SLN gels had pH values ranging from 5.20 to 6.30. There were no visible clumps or signs of phase separation in the gel, which demonstrated homogeneous consistency.

#### 2.2.2. Rheological Studies

The rheograms of all SCZ-SLNs gels revealed non-Newtonian flow behavior with decreased viscosity by the constant increase in the shear rate (Figure 6). Rheological analyses of all the SCZ-SLN gels showed the inverse relationship between the shear rate and viscosity of all the SCZ-SLN gels, which is required for their proper topical application. The shear rate versus viscosity graph is illustrated in Figure 6.

#### 2.2.3. Spreadability and Extrudability

All SCZ-SLN-based gels were easily spreadable and extrudable. All formulations showed better gel strength. Spreadability determines the ease of applying a gel to the skin’s surface. All fabricated formulations showed spreadability from 28.4 ± 0.48 to 35.2 ± 0.91 g cm/s. The values of extrudability were in the range of 10.43 ± 0.25 to 13.67 ± 0.12 g/cm^2^.

### 2.3. In Vitro Drug Release and Drug Release Kinetics

The graph was plotted between time and % drug release as shown in Figure 7. The formulation from SLN3 to SLN5 showed an inverse relationship with drug release; as the concentration of GMS was increased from 3% to 5%, the % of drug released decreased. However, the formulation containing 2% GMS (SLN2) showed an upward trend from 14.30 to 85.29% in 12 h. In contrast, the formulation containing 1% GMS (SLN1) depicted a % drug release of 81.29%. The drug solution in DMSO showed 38.29% SCZ released in 12 h.

The outcomes of the in vitro drug release were correlated with several kinetic models to comprehend the mechanism and kinetics of drug release. According to the release kinetics, the SLN2 optimized formulation followed first-order kinetics, as shown by the greatest R^2^ of 0.98 in Table 3. The values of ‘n’ were 0.53 to 0.57 suggesting the non-Fickian (anomalous) diffusion release mechanism.

### 2.4. Ex Vivo Skin Permeation

To assess the permeability of the SCZ-SLN2 gel and drug-loaded gel formulations, a graph of % drug permeation vs. time was created, as shown in Figure 8. Due to the lipid content of SLNs, the SCZ-SLN2 gel demonstrated a higher flux value than drug-loaded gel formulations. As indicated in Table 4, the enhancement ratios of the SCZ-SLN2 gel were also found to be 2.62, indicating an increased rate of drug permeation through the skin via the SLN gel. The present study compared the permeation of the SLN gel with a control gel and revealed a significant difference (*p* < 0.05) in the statistical analysis.

### 2.5. Stability Studies

As indicated in Table 5, the stability assessment of the optimized freeze-dried SCZ-SLNs revealed a minor decrease in the drug content from 94.18 to 92.32% at ambient temperature and from 94.11 to 93.07% at freezing (−20 ± 5 °C). During the stability analysis, the nanoparticles’ size increased over three months, from 89.81 nm to 106.5 nm at room temperature.

The drug content of the fabricated SCZ-SLN gel decreased over a 3-month observation period. As indicated in Table 6, the drug concentration was discovered to be 90.40 and 92.07% at room and freezing temperatures, respectively, after three months. It suggested that the preparation is more stable when chilled than at room temperature. It was observed that the pH of the SCZ-SLN gel slightly increased with time from 5.6 to 6.4.

### 2.6. Antifungal Activity

A zone of inhibition (ZOI) experiment was used to assess the SCZ-SLNs’ antifungal activity against *Candida albicans* and *Trichophyton rubrum*. According to Table 7, the ZOIs of the SCZ-SLNs, the SCZ-SLN gel and the sulconazole gel were 24.36, 26.5 and 18.4 mm, respectively, against *Candida albicans* after treatment for 48 h, and those against *Trichophyton rubrum* were 21.2, 16.8 and 17.3 mm, respectively. The lack of activity against *Candida albicans* or *Trichophyton rubrum* in the blank SLN formulation suggested that the SCZ-SLN formulation lacked innate antifungal effectiveness. The SCZ-SLNs and SCZ-SLN gel had higher antifungal action against *Candida albicans* than the SCZ carbopol-based gel. The SCZ-SLNs and SCZ-SLN gel outperformed the MCZ cream regarding antifungal activity against *Trichophyton rubrum*. Therefore, compared to the SCZ carbopol-based gel, the SCZ-SLN gel was more potent against *Candida albicans* and *Trichophyton rubrum*. It showed that the SCZ-SLNs, which are nanoscale drug carriers, had high antifungal potential. Due to the SCZ-SLNS’s substantial specific surface area, it could easily interface with the fungal cells’ surfaces and display antifungal properties. The capacity of the SCZ-SLNs to approach the ergosterol content of fungal hyphae may be responsible for their antifungal activities. The previously reported ZOIs of SCZ-fabricated nanoemulsions against *Trichophyton rubrum* and *Candida albicans* were 23.5 and 20.4 mm [22].

### 2.7. In Vivo Study

Lesions and wounds began to heal in those treated with SCZ-SLN2 (B group), while bleeding started in the wounds of the animals treated with the blank gel (A group). The therapeutic effect of the SCZ-SLN gel was slightly greater than the SCZ carbopol-based gel. Animals treated with the SCZ carbopol-based gel were not completely recovered from the infection on day 12, as illustrated in Figure 9 (C group).

### 2.8. Histopathology

The negative control group (to which fungus was not induced) showed the normal architectural structure of the epidermal and dermal layers of the skin. No signs of inflammatory response were observed (Figure 10A).

The animals treated with only blank gel showed a compact layer of hyperkeratosis with focal acanthosis. Fungal hyphae were also present in the epidermal layer with focal interface dermatitis. The dermal layer also showed dense chronic inflammation (Figure 10B).

The skin structure of the group treated with the optimized SCZ-SLN2 gel showed normal dermal and epidermal layers of the skin structure. Focal acanthosis and hyperkeratosis characters were absent. At the plane interface, focal dermatitis was absent (Figure 10C).

## 3. Discussion

In the current investigation, we aimed to create SCZ SLNs that had been lyophilized and subsequently converted into gel using CP 934. The surfactant utilized in the preparation is one of the elements impacting the particle size of SLNs. The surfactant lowers the interfacial tension among the lipid and aqueous phases, resulting in small-size nanoparticles. A steric barrier on the particle surface, which prevents nanoparticles from aggregating, also provides stability to the SLNs. Analysis was conducted of the preliminary parameters for both SLNs and SLN gel formulations. Our findings showed that the amount of GMS as a lipid directly correlates with the size of the particles. Similar findings were interpreted by Sooho Yeo et al. [23]. The optimized SCZ-SLN2 zeta potential values were discovered to be −26.98 mV. The phosphate groups in Phospholipon^®^ 90 H can be attributed to the negative charge on the SLNs. Phospholipon^®^ 90 H would display a negative charge in water at neutral pH [24]. As a result, the drug–phospholipid complex is dynamically stable due to the consequence of electrostatic repulsion induced by the overall negative charge, which also raises the possibility that SCZ is physically stable within SLNs. Theoretically, the PDI ranges from 0 to 1, and values greater than 0.5 signify a wide particle size distribution [19]. The PDI measures the homogeneity degree of the dispersion. A good particle dispersion was observed in the current study since the PDI was less than 0.5.

The results showed that the EE (%) of SLN formulations continued to decline when the GMS concentration was gradually raised from 2 to 5% *w*/*v*. The partition phenomena could explain the observed reduction in EE. High GMS levels in the external phase may enhance the partition of the drug from the internal to the external phase due to the improved solubilization of the SCZ in the external aqueous phase, which permits the diffusion and dissolution of the drug [25]. TEM images of SLNs also give insight into the size, shape or structure of the particles. TEM revealed that the SCZ-SLNs were often uniformly shaped and rounded in appearance. However, it is important to note that the sizes obtained by TEM were very different from those obtained by the zeta sizer-based light scattering technique. The apparent hydrodynamic size, which considers the aqueous layers encompassing the SLNs, is determined by the particle size as determined by the zetasizer [26].

The FTIR results revealed a weak intermolecular interaction, whereas sharp exothermic peaks in the formulation were observed through DSC/TGA analysis. Solid-state characterization was also studied by DSC. Since most drugs in BCS class II display crystallinity, which is a significant barrier to drug solubility and bioavailability, DSC curves of pure SCZ also showed the molecule to be crystalline. The endothermic peak in Phospholipon^®^ 90 H displayed a glass transition temperature and phase transition from a gel phase to liquid crystalline phase. The GMS also showed a sharp endothermic peak. By contrast, the sharp endothermic peak of GMS overlapped with the lipid endothermic peak in the SCZ-SLN formulation, demonstrating how SCZ transforms when combined with lipids and surfactants. However, the absence of a comparable endothermic SCZ peak in the SCZ-SLNs demonstrated that the drug had changed from its crystalline form to an amorphous state, suggesting an improved solubility and bioavailability. No new peaks were observed in the DSC/TGA thermograms of SCZ-SLNs, demonstrating the chemical compatibility, stability and additive behavior between the formulation components. This study showed how an SLN system could improve the drug’s dissolution and drug release from SLNs. PXRD investigations were used to validate these findings further. The DSC results complemented the PXRD results and the crystalline structures of SCZ and Phospholipon^®^ 90 H.

Furthermore, the primary representative peaks of SCZ, Phospholipon^®^ 90 H and GMS on the PXRD graph indicated the presence of crystals. However, no such peak was present in SCZ-SLNs, supporting the amorphization of SCZ from its crystalline state. Both of these findings supported the idea that the SLNs might alter the physical state of the SCZ, enhancing the solubilization and raising bioavailability [27].

All formulations displayed pseudoplastic non-Newtonian flow in the viscosity analysis, and the viscosity showed an inverse relationship with shear rate. The spreadability results demonstrated that the formulation spread easily over the skin. The SCZ-SLNs have a pH between 5.2 and 6.3, which lies in the normal range. It was seen that the release began quickly and then slowed down. This initial rapid release rate can be caused by the drug desorption associated with SLNs surfaces, and the subsequent slower release can be explained by the fact that solubilized drugs can only be released gradually from lipid matrices by dissolution and diffusion. In formulations SLN2 to SLN5, as the concentration of GMS was raised, the drug release from the SCZ-SLNs was consequently reduced. The increased lipid content (GMS) in the SLN preparation increased the lipid shell’s thickness, increasing the length of the diffusion path and lowering the drug release. Our results are consistent with a prior study which found that lower drug release was caused by a higher concentration of GMS [28]. The ex vivo study illustrated a higher SCZ-SLN gel flux value due to the lipid content of SLNs compared to the control SCZ gel and had an almost three-fold increase in enhancement ratio. The stability showed that freeze-dried SCZ-SLNs revealed a minor decrease in the drug content in the refrigerated storage. Although, the particle size of SLNs increased over three months at room temperature, where leaching of drugs from the SCZ-SLNs can be a potential explanation. At the freezing storage conditions, this leaching was less than at ambient temperature. During the stability study, the drug content of the fabricated SCZ-SLN gel decreased during a 3-month observation period. The pH of the gel showed a slight change with time. These results also agreed with those reported by Rudhrabatla et al. on the stability of melphalan SLNs with stealth characteristics [29]. Batool et al. also reported comparable results with Carbopol gel against leishmaniasis [30].

The antifungal activity against *Candida albicans* and *Trichophyton rubrum* demonstrated a higher zone of inhibition than the control marketed SCZ carbopol-based gel. Thus, it exhibits higher antifungal potential against topical fungal infection. The fabricated SLNs are safe and effective drug delivery vehicles for improved oral and topical delivery of antifungal drugs, according to the acute oral toxicity investigation results.

## 4. Materials and Methods

### 4.1. Materials

SCZ was purchased from BioTek, Tigan Street, Highland Park, Winooski, VT, USA, and GMS was obtained as a gift by IOI Oleochemicals, Germany. Tween 20, 1640 medium and dimethyl sulfoxide (DMSO) were purchased from Sigma Aldrich, Saint Louis, CA, USA. Phospholipon^®^ 90 H was received as a gift sample by Lipoid GmbH, Frigenstr., Ludwigshafen, Germany. Double-distilled water was freshly prepared in the lab. 

### 4.2. Preparation of SLNss

SLN dispersions with different lipid concentrations were prepared by the high shear homogenization method (HSHM) as Neeraj Kumar et al. reported with slight modification [31]. SCZ was added in GMS previously melted at 75 °C using a water bath to prepare the lipid phase. The aqueous phase was fabricated by mixing the surfactant in double-distilled water at 75 °C. The hot lipid phase was dispersed slowly in the aqueous phase under constant stirring using a hot plate and stirrer (Eisco, Scientific, 788 Old Dutch Rd, Victor, NY, USA) at the same temperature. The mixture was homogenized by a high shear homogenizer (Polytron PT 1200E, Kinematica AG, Lucerne, Switzerland) at 20,000 rpm for 5 min. The mixture was stirred and cooled to room temperature to obtain SLN dispersion. Sonication was performed for 10 min to obtain fine SCZ-SLNs. The dispersions were frozen at −75 °C in an ultralow temperature freezer (DW-HL398/SA, Meling Biomedical, Zishi Road, HeFei City, AnHui, China) and lyophilized in a freeze dryer (Lyovapor L-200, Buchi, Flawil, Switzerland) for 48 h to obtain SCZ-SLNs dry powder and stored at room temperature in a vacuum desiccator for further characterization. The composition of SLN formulations is shown in Table 8**.**

### 4.3. Characterization of SCZ-SLNs

#### 4.3.1. Particle Size, Polydispersity Index (PDI) and Zeta Potential

The dynamic light scattering technique was used to determine the size of the nanoparticles and PDI by changing the intensity of light scattered by Zeta sizer (ZS-90, Malvern, UK). Samples were diluted with filtered double-distilled water (1:100) and were filled in a disposable cuvette for analysis at constant temperature (25 °C), refractive index, viscosity and dielectric constant. Zeta potential and surface charges were also measured by Zeta sizer; after dilution, the samples were kept in Zeta cells at 25 °C.

#### 4.3.2. FTIR Spectroscopy

SCZ, GMS, tween 20, Phospholipon^®^ 90 H and the formulations were analyzed by FTIR spectrophotometer (Bruker, Tensor 27, Rheinstetten, Germany) to find the interaction among the components of the SLNs. After taking a sample on ATR crystal, the face of the crystal was pressed and scanned for 16 s in the range of 4000 to 500 cm^−1^.

#### 4.3.3. Transmission Electron Microscopy (TEM)

A transmission electron microscopic study determined the shape of the SCZ-SLNs’ dispersion (JEM 1230, JEOL, Tokyo, Japan). SCZ-SLNs’ dispersion was diluted with deionized water and vortexed. The diluted sample was stained with 2% PTA (tungstic phosphate acid) solution, and one drop of the stained diluted sample was put on the carbon-coated copper grid and dried at room temperature before the examination. A voltage of 200 KV was required to obtain TEM images.

#### 4.3.4. Thermal Analysis

Differential scanning calorimetry (DSC) of SCZ, GMS, Phospholipon^®^ 90 H and SLN2 formulation was carried out for thermal analysis on a calorimeter (SDT Q600, V8.3 Build 101, Luken Drive, New Castle, DE, USA). An empty reference aluminum pan was used to check the heat flow. Under constant purging of nitrogen gas and at a 10 °C/min heating rate, accurately weighed samples were taken on aluminum pans and scanned at 30 °C to 250 °C for thermal analysis, and thermograms of the samples were recorded.

Thermogravimetric analysis (TGA) was performed to determine the thermal decomposition of GMS, SCZ, Phospholipon^®^ 90 H and SLN2 formulation by using a thermogravimetric analyzer (SDT Q600, V8.3 Build 101, Luken Drive, New Castle, DE, USA); 2 mg sample was put in an aluminum pan, and TGA analysis was carried out at a temperature ranging from 25 to 500 °C, with a heating rate of 10 °C/min and nitrogen flow rates of 20 mL/min.

#### 4.3.5. X-ray Diffraction Analysis (XRD)

The crystalline behavior of SCZ, GMS, Phospholipon^®^ 90 H and SCZ-SLNs was determined by X-ray diffractometer (JDX-3532, JEOL, Tokyo, Japan). The samples were kept in the holder at 40 kV operating voltage and 35 A current, while the X-ray radiation source was Cu Kα (λ = 1.54050 Å). The sample was scanned at 0.1° diffracting angle with 2*θ* and 1 s of counting time. The diffractograms of all the samples were recorded [32].

#### 4.3.6. Encapsulation Efficiency (EE)

Encapsulation efficiency of SCZ-SLNs was determined by the indirect method by calculating the amount of unentrapped drug; 1 mL of SCZ-SLNs dispersion was taken, and centrifugation was carried out at 12,000 rpm for 60 min in an eppendorf tube, SLNs pellets were separated. The supernatant was diluted, and the amount of unentrapped SCZ was measured by analyzing the diluted sample spectrophotometrically at 229 nm. The EE (%) was measured according to the Equation.
(1)$$\mathrm{EE}\;\left(\%\right)=\frac{\mathrm{Total}\;\mathrm{amount}\;\mathrm{of}\;\mathrm{SCZ}-\mathrm{free}\;\mathrm{unentrapped}\;\mathrm{SCZ}\;\mathrm{amount}}{\;\mathrm{total}\;\mathrm{amount}\;\mathrm{of}\;\mathrm{SCZ}}\times 100$$

#### 4.3.7. % Production Yield

To obtain the % production yield, the % weight of the final formulation after drying was determined and compared with the initial total amount of all materials used to prepare the formulations [33]. The% production yield was calculated from Equation (2).
(2)$$\%\;\mathrm{production}\;\mathrm{yield}\;=\frac{\mathrm{Actual}\;\mathrm{production}\;\mathrm{yield}}{\mathrm{Theoretical}\;\mathrm{yield}}\times 100$$

### 4.4. Preparation of SCZ-SLN Gel

SCZ-SLNs were converted into a gel carrier system by using carbopol (CP 934), where 0.75% *w*/*v* CP 934 was soaked in distilled water for 24 h and 0.2% *w*/*v* methylparaben was added as a preservative. SLNs’ dispersion, CP 934 and methylparaben solutions were stirred for 10 min at 1500 rpm. The measured amount of SCZ-SLNs was slowly added to the gel system, mixed for 10 min, and neutralized with a few drops of triethanolamine until pH 6.5. To remove entrapped air, the prepared gels were allowed to stand for 24 h. The formed gels were kept in suitable containers for further studies. 

### 4.5. Characterization of SCZ-SLNs Gel

#### 4.5.1. pH, Homogeneity and Grittiness

For homogeneity, appearance and presence of any clog, the fabricated SCZ-SLN gel was examined visually. Using a digital pH meter (JENWAY 350, Beacon Road Stone Staffordshire, UK), the pH of the gel was noted. The gel formulation was examined under a light microscope for the presence of any foreign material.

#### 4.5.2. Rheological Studies

Rheological properties were studied by means of viscometer (Brookfield DV-III, Ribbleton, Preston PR2 6UE, Preston, CT, USA) by employing spindle number 64 at different shear rates, applying varying torque values to the gel. The measurements for each sample were taken across a range of shear rates from 2 to 100 s^−1^.

#### 4.5.3. Spreadability and Extrudability

The spreadability of the prepared gel was determined by using two glass slides with wooden box apparatus. For the gel to spread well, the two slides should be separated in the shortest amount of time and with the least force. A 95 g SCZ-SLNs gel sample was placed on top of one slide, and the second slide was placed over the gel. The gel spreadability was determined by measuring the area covered by the gel. Spreadability was measured by Equation (3).
(3)$$\mathrm{Sp}\;=W\;\times \frac{L}{t}$$

Sp is the spreadability in g cm/s, W is the weight of gel placed on an upper slide, L represents glass slide length and t is the time taken to separate slides [34].

20 g of SLNs gel was packed in a collapsible tube; the tube was closed with a cap and pressed on the other end. The cap was removed, and the SCZ-SLNs gel was extruded. The weight of extruded gel was estimated [35].

#### 4.5.4. Drug Content

The SCZ-SLNs gel was accurately weighed, diluted with methanol and sonicated for 45 min. Then, 5 mL was pipetted out after sonication and again diluted with methanol, this time up to 50 mL, and the absorbance was measured by uv/vis spectrophotometer at 229 nm [36].

### 4.6. In Vitro Drug Release and Drug Release Kinetics

In vitro release was carried out to determine the % release of SCZ from SLNs. Before the drug release experiment, the cellophane dialysis membrane (12,000 to 14,000 Da) was submerged in the phosphate-buffered saline (PBS), pH 7.4, overnight. SCZ-SLN formulation was kept in the cellophane dialysis membrane in USP type II dissolution apparatus (Pharma test W00 4895, Hainberg, Germany), and both ends of the membrane were fastened tightly with threads to prevent any leakage. The membrane was placed in 900 mL of PBS pH 7.4. At the pre-determined time interval, 3 mL of the buffer media sample from the basket was withdrawn and replaced immediately with 3 mL of freshly prepared buffer media. The samples were filtered, appropriately diluted and analyzed by uv/vis spectrophotometer (PerkinElmer Lambda 35, Winter Street, Waltham, MA, USA) at 229 nm to determine the % drug released at different time intervals from the SCZ-SLNs gel formulations.

The DDSolver.xl, Add-In Excel program was used to measure the various parameters of the kinetic release models. Based on the correlation coefficient (R^2^) values, the best-fit model was chosen, and the release mechanism was studied.

### 4.7. Ex Vivo Skin Permeation

The ex vivo permeation of SCZ-SLN gel and SCZ gel containing CP 934 (control formulation) was conducted on the abdomen skin of albino rats using the previously described method by Mudassir et al. Using a shaving razor, the hair and subcutaneous tissues were eliminated. After washing with distilled water and removing any remaining fat material, the dermis side was kept in PBS pH 7.4 until it was needed. A Franz diffusion cell (PERME Gear, Inc. No: 4G-01-00-15-12, Riegelsville, PA, USA) with surface area of 1.76 cm^2^ having 12 mL receptor compartment volume was employed. The receptor compartment was filled with pH 7.4 PBS at 100 rpm and a temperature of 37 ± 0.5 °C, and the dermal area of the skin was mounted below the donor compartment. The sample was placed in the donor compartment when the temperature was maintained. At predefined intervals (0.5 to 12 h), the sample solution was removed from the receptor compartment and replaced with PBS. Finally, the absorption of samples was measured by uv/vis spectrophotometer (PerkinElmer Lambda 35, Riegelsville, PA, USA) at 229 nm, and different parameters such as permeation flux and enhancement ratio were calculated using Equations (4)–(6) [37].
(4)$$J_{P}=k_{P}\;\times {\;C}_{i}$$
(5)$$k_{P}={\;V}_{R\;}\times \frac{C_{R\;}}{A\;\times \;T}{\;\times \;C}_{D}$$
(6)$$\mathrm{Enhancement}\;\mathrm{ratio}\;=\frac{J_{P}\;\left(\mathrm{formulation}\right)}{J_{P}\;\left(\mathrm{control}\right)}$$

‘J_p_’ donates flux, ‘k_p_’ demonstrates coefficient of permeability, ‘C_i_’ is initial drug concentration, ‘C_R_’ and ‘C_D_’ depicts the concentration of the drug in the receptor and donor compartment, ‘A’ is the surface area of Franz diffusion cell and ‘T’ is time.

### 4.8. Antifungal Activity

To evaluate the antifungal efficacy of the SCZ-SLNs, SCZ-SLN gel, sulconazole gel and blank SLN formulation, zones of inhibition (ZOI) were assessed against *Candida albicans* and *Trichophyton rubrum*. The SCZ carbopol-based gel served as a positive control. Sample concentrations ranged from 0.48 to 250 μg/mL, and inoculum suspensions of *Candida albicans* and *Trichophyton rubrum* were diluted with RPMI-1640 to concentrations of 0.5–2.5 × 10^3^ and 2.4 × 10^4^ colony-forming units per mL, for *Candida albicans* and *Trichophyton rubrum.* Then, a Sabouraud agar plate of 9 cm was uniformly infected with 100 μL of inoculum suspensions. The samples were put into a well with a 6 mm diameter. The plates were then incubated for 48 h at 35 ± 2 °C. Based on the area where fungal growth is inhibited surrounding the treatment solution, the antifungal effectiveness was evaluated by measuring ZOI in mm [38].

### 4.9. In Vivo Study

The study was conducted on male rabbits divided into three groups (n = 3) to assess the in vivo antifungal activity of formulated SCZ-SLN gel. Group 1 received blank gel. Group 2 received SLN2 gel formulation, while Group 3 received SLN carbopol-based gel formulation. Each group of rabbits was kept in separate cages with a standard diet. Nine rabbits were infected using *Candida albicans* as the infectious agent. *Candida albicans* were received from the laboratory of the Department of Biotechnology, Bahauddin Zakariya University Multan, Pakistan. The collected fungi were kept alive on nutrient agar plates and refrigerated at 4 °C. In order to resuscitate the organisms, they were afresh cultivated before being applied to the animals. Each rabbit had its dorsal region depilated. The rabbits’ depilated skin was exposed to a freshly cultured infectious inoculum of *Candida albicans*, which was kept on for three days to produce the initial signs of active infection (redness and scales). Each formulation was administered to the animals once daily (OD) beginning on day 4 of the study. The therapy was maintained until the infection had completely cleared up (12 days). In vivo studies were carried out according to ethical guidelines approved by the ethical committee of the faculty of Pharmacy Bahauddin Zakariya University Multan, Pakistan, vide reference NO.01/PEC/2022 dated 5 June 2022.

### 4.10. Histopathology Study

Histopathology studies were carried out by obtaining fresh samples from the skin of rabbit’s back, fixation, dehydration, clearing, wax infiltration, embedding or blocking out and microtomy. Tissue slides were stained using Hematoxylin and Eosin H and E (FD NeuroTechnologies, Inc. Guilford Road, Suite, Columbia), followed by examination using a compound microscope.

### 4.11. Stability Studies

Following the International Conference on Harmonization (Q1A-R2) recommendations (FDA, 2003), the freeze-dried SCZ-SLNs and SCZ-SLN gel were kept in air-tight containers. To monitor the stability of the SCZ-SLNs, the particle size and the % of drug content were assessed. Additionally, a three-month accelerated stability investigation for the SCZ-SLN gel was conducted. Initially, a collapsible aluminum tube containing the SCZ-SLN gel was enclosed and maintained at 25 ± 2 °C with a 60 ± 2% humidity level. Similarly, samples were also stored at a lower temperature (−20 ± 5 °C). The pH and drug content of the SCZ-SLN gel were measured and determined after a predefined amount of time [39].

### 4.12. Statistical Analysis

The data were statistically examined to measure the level of significance. Data are presented as the mean and standard deviation (SD) of at least three samples. The ex vivo permeation study was examined using analysis of variance (ANOVA) with the Tukey test by GraphPad Prism 8.0.1. *p* < 0.05 was chosen as the cutoff for statistical significance.

## 5. Conclusions

In order to reduce the dosage, dosing schedule and adverse effects connected with oral medication, the topical SCZ-SLN formulation was fabricated and optimized by using GMS and Phospholipon^®^ 90 H as a lipid matrix and tween 20 as a surfactant by high-pressure homogenization process. The drug to phospholipid ratio was tuned to achieve the desired particle size with the maximum % encapsulation efficiency. This study presents an optimized SLN formulation with enhanced permeability, controlled release and fewer side effects as a substitute to conventional preparations for transdermal fungal therapy. The SCZ-SLN gel displayed better skin deposition and in vitro and in vivo antifungal activity in comparison to the commercial formulation. The safety, accessibility and tolerability of SCZ were enhanced by this improved SCZ-SLN gel. In summary, SCZ-SLNs are safe, efficient and affordable to fulfill the need for therapeutic agent administration via oral and transdermal routes. The current investigation can provide supporting evidence for further clinical development of SLN-based gels as the next generation of topical anti-infectives.

## Figures and Tables

**Figure 1 molecules-28-07508-f001:**
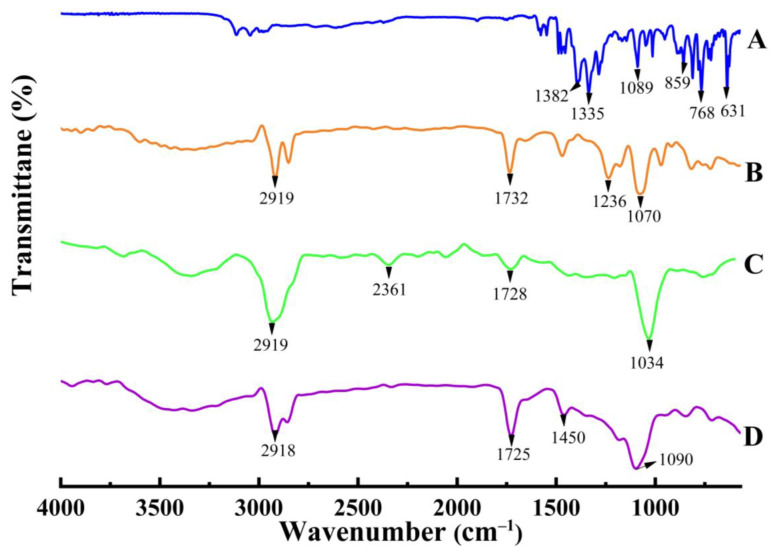
FTIR of Sulconazole (**A**), Phospholipon^®^ 90 H (**B**), GMS (**C**) and SLN2 formulation (**D**).

**Figure 2 molecules-28-07508-f002:**
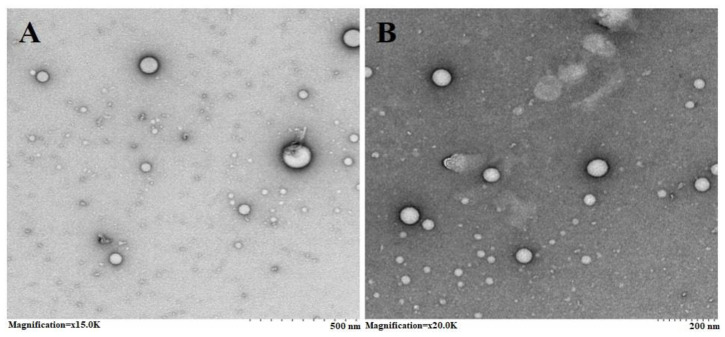
TEM image of SLN2 at ×15 k (**A**) and ×20 k (**B**) magnifications.

**Figure 3 molecules-28-07508-f003:**
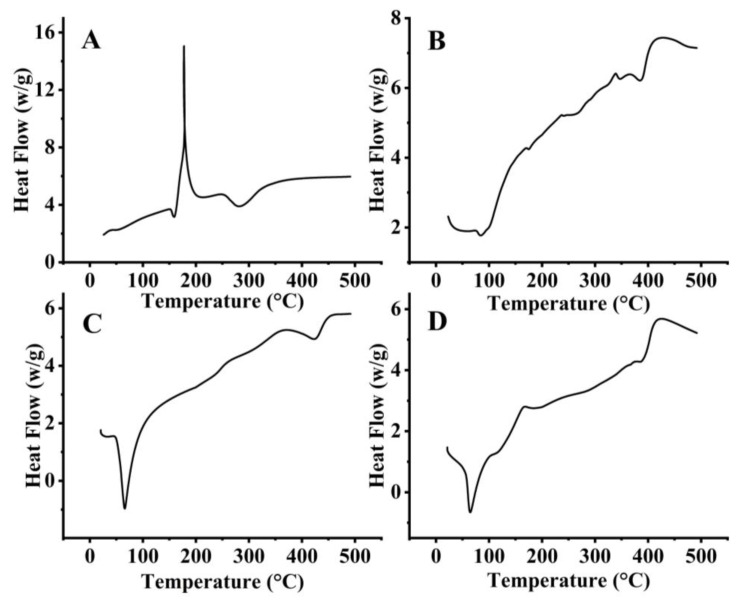
DSC of Sulconazole (SCZ) (**A**), Phospholipon^®^ 90 H (**B**), GMS (**C**) and SLN2 (**D**).

**Figure 4 molecules-28-07508-f004:**
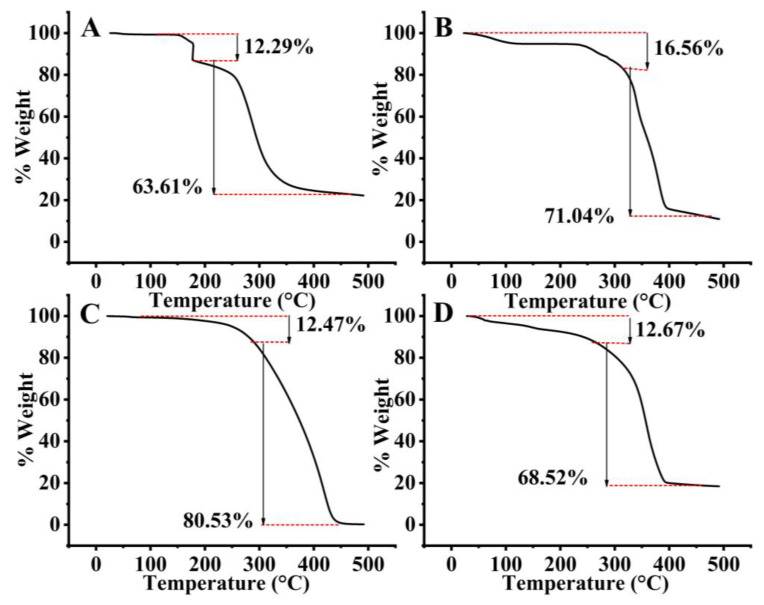
TGA of Sulconazole (**A**), Phospholipon^®^ 90 H (**B**), GMS (**C**) and SLN2 (**D**).

**Figure 5 molecules-28-07508-f005:**
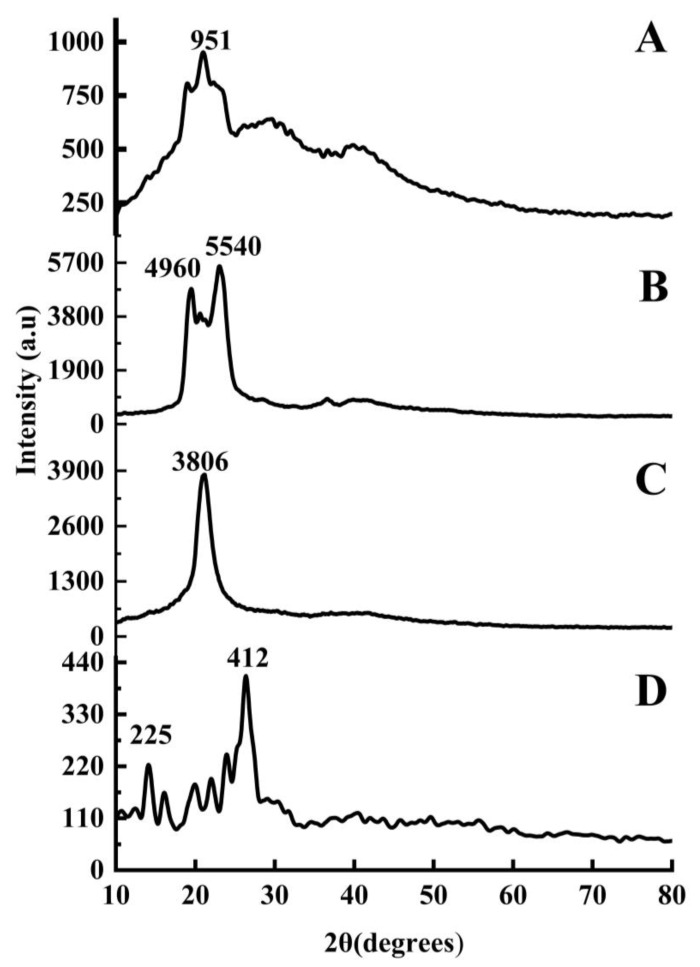
SLN2 (**A**), GMS (**B**), Phospholipon^®^ 90 H (**C**) and Sulconazole SCZ (**D**).

**Figure 6 molecules-28-07508-f006:**
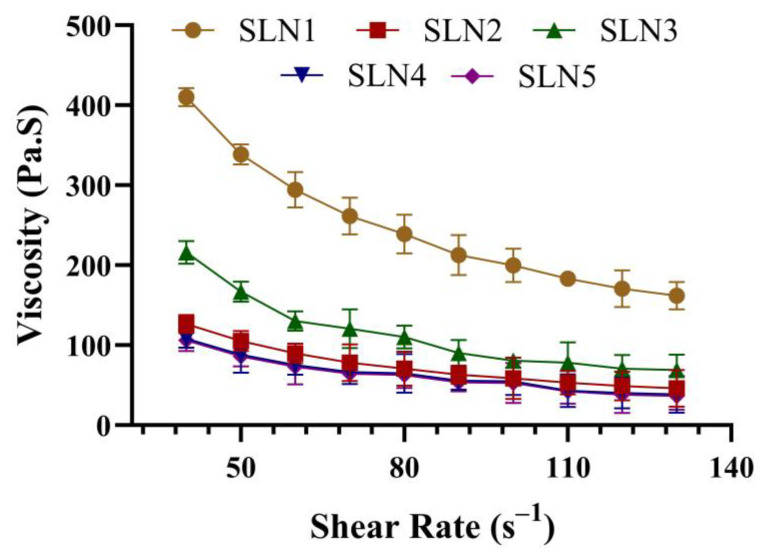
Rheograms of fabricated formulations (Mean ± SD, n = 3).

**Figure 7 molecules-28-07508-f007:**
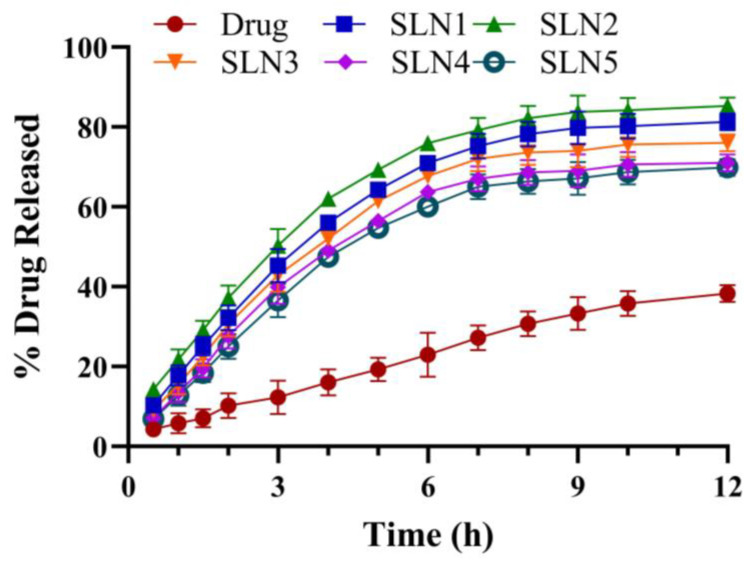
Percent drug release of all formulations in 12 h (Mean ± SD, n = 3).

**Figure 8 molecules-28-07508-f008:**
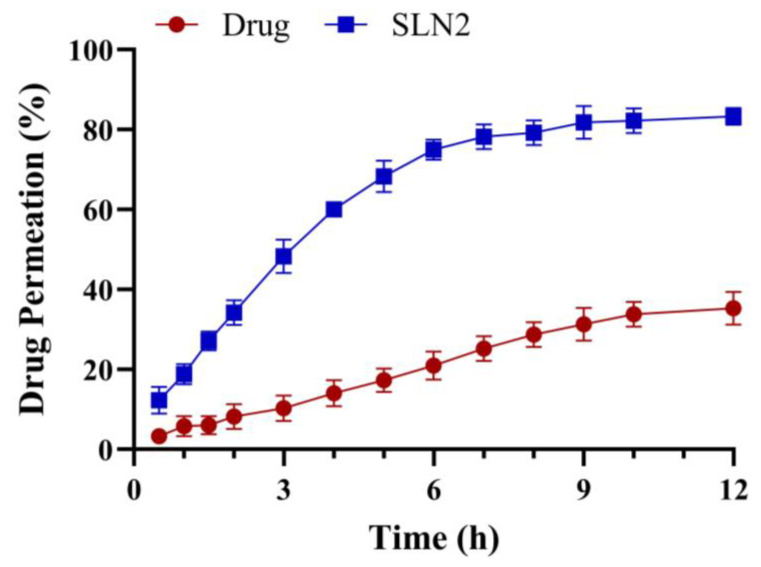
Ex vivo skin permeation of solid lipid nanoparticle gel formulation (SLN2) and pure drug as control (Mean ± SD, n = 3).

**Figure 9 molecules-28-07508-f009:**
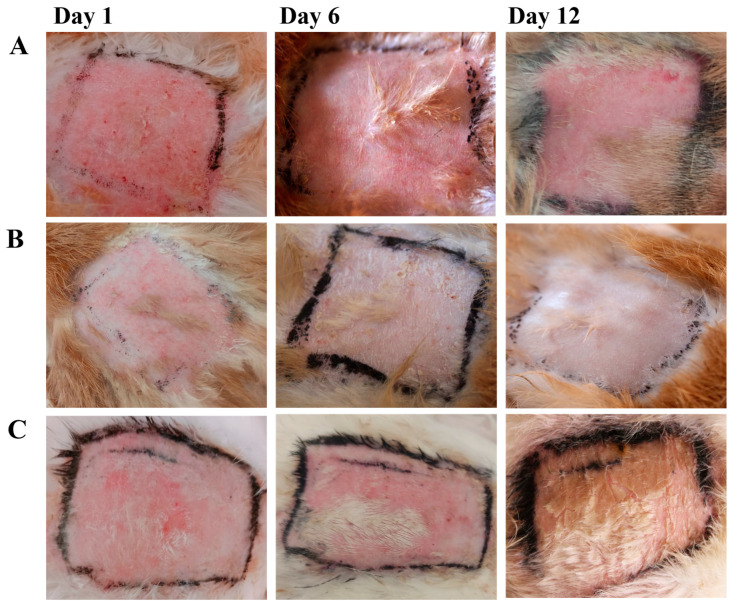
In vivo study of blank gel (**A**), SLN2 gel (**B**) and sulconazole carbopol-based gel (**C**) at Day 1, Day 6 and Day 12.

**Figure 10 molecules-28-07508-f010:**
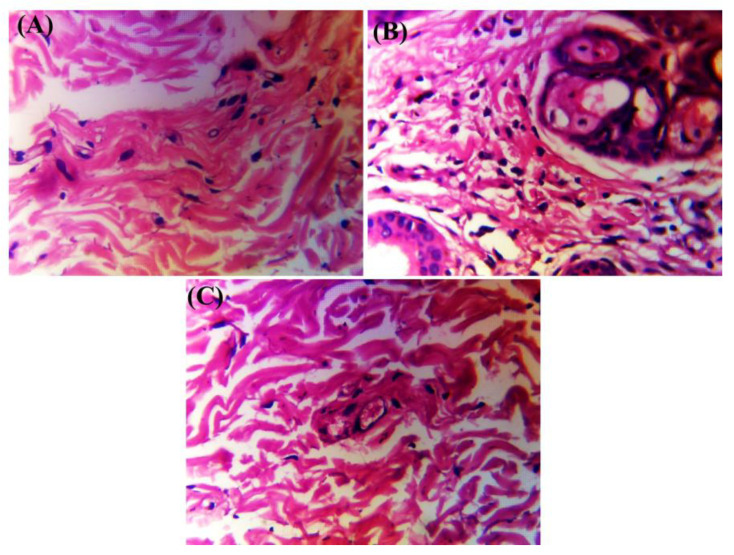
Histopathology examination of negative control group (**A**), blank treatment skin (**B**) and SCZ-SLN2 gel (**C**) treatment skin.

**Table 1 molecules-28-07508-t001:** Particle size, PDI and ZP of formulations (Mean ± SD, n = 3).

Formulation Code	Particle Size	PDI	Zeta Potential (mV)
SLN1	60.31 ± 1.33	0.266 ± 0.03	−28.52 ± 1.23
SLN2	89.81 ± 2.64	0.311 ± 0.07	−26.98 ± 1.19
SLN3	126.32 ± 4.24	0.378 ± 0.02	−19.44 ± 1.63
SLN4	178.93 ± 3.55	0.406 ± 0.10	−16.40 ± 1.08
SLN5	194.9 ± 2.26	0.304 ± 0.13	−21.14 ± 2.98

**Table 2 molecules-28-07508-t002:** The EE (%) and production yield (%) of fabricated formulations (Mean ± SD, n = 3).

Formulation Code	EE (%)	Production Yield (%)
SLN1	83.33 ± 1.45	88.33 ± 2.75
SLN2	86.52 ± 0.53	91.35 ± 1.48
SLN3	81.89 ± 1.67	85.89 ± 2.38
SLN4	79.73 ± 2.45	83.73 ± 1.92
SLN5	77.16 ± 1.42	80.16 ± 1.15

**Table 3 molecules-28-07508-t003:** Kinetic release models of Sulconazole SLN gel formulations.

Formulation Code	Zero-Order		First Order		Higuchi Model		Korsmeyer-Peppas Model		
	K_o_	R^2^	K_1_	R^2^	K_H_	R^2^	K_KP_	n	R^2^
SLN1	9.18	0.65	0.19	0.98	26.2	0.94	24.5	0.53	0.94
SLN2	9.74	0.53	0.22	0.98	27.9	0.94	28.4	0.49	0.94
SLN3	8.64	0.65	0.16	0.96	24.6	0.93	22.8	0.53	0.93
SLN4	9.05	0.68	0.14	0.96	22.9	0.92	20.6	0.55	0.93
SLN5	7.79	0.72	0.13	0.96	22.0	0.93	19.1	0.57	0.93

**Table 4 molecules-28-07508-t004:** Permeability study of SLNs (Mean ± SD, n = 3).

Formulation Code	% J (μg cm^−2^ h^−1^)	K_p_ (cm h^−1^)	Enhancement Ratio
Control SCZ gel	35.29 ± 3.14	4.61 ± 2.40	2.62
SCZ-SLN2 gel	83.60 ± 2.95	6.57 ± 1.38	

**Table 5 molecules-28-07508-t005:** Freeze-dried Sulconazole SLN2 formulation stability studies (Mean ± SD, n = 3).

	Drug Content		Particle Size (nm)
	Room Temperature	Refrigerator Temperature	Room Temperature
Initial	94.18 ± 0.46	94.11 ± 0.46	89.81 ± 2.64
After 1 month	93.53 ± 0.52	94.08 ± 0.09	97.03 ± 3.42
After 2 months	93.10 ± 0.74	93.25 ± 0.62	103.8 ± 5.21
After 3 months	92.32 ± 0.95	93.07 ± 0.75	106.5 ± 4.24

**Table 6 molecules-28-07508-t006:** Sulconazole SLN2 gel stability studies (Mean ± SD, n = 3).

	Drug Content		pH
	Room Temperature	Refrigerator Temperature	Room Temperature
Initial	94.18 ± 0.25	94.18 ± 0.32	5.6 ± 0.75
After 1 month	93.87 ± 0.90	93.08 ± 0.46	5.8 ± 0.18
After 2 months	92.06 ± 0.73	92.25 ± 0.63	6.4 ± 0.65
After 3 months	90.40 ± 0.52	92.07 ± 0.45	5.7 ± 0.08

**Table 7 molecules-28-07508-t007:** ZOI results against *Candida albicans* and *Trichophyton rubrum* (Mean ± SD, n = 3).

Formulation Code	Zone of Inhibition (ZOI)	
	*Candida albicans*	*Trichophyton rubrum*
SCZ-SLNs	24.3 ± 1.90	21.2 ± 2.83
SCZ-SLNs gel	26.5 ± 3.42	23.7 ± 1.45
SCZ-gel	17.1 ± 1.43	16.4 ± 3.5
Blank SLNs	00	00

**Table 8 molecules-28-07508-t008:** Formulation table of sulconazole solid lipid nanoparticles.

Formulation Code	SCZ	GMS	Tween 20	Phospholipon^®^ 90 H
SLN1	1%	1%	2%	0.75%
SLN2	1%	2%	2%	0.75%
SLN3	1%	3%	2%	0.75%
SLN4	1%	4%	2%	0.75%
SLN5	1%	5%	2%	0.75%

## Data Availability

Data are contained within the article.

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
