# Peer review of "Sulconazole-Loaded Solid Lipid Nanoparticles for Enhanced Antifungal Activity: In Vitro and In Vivo Approach"

_molecules, 2023, doi:10.3390/molecules28227508_

Round 1

Reviewer 1 Report

Comments and Suggestions for Authors

Authors synthesized sulconazole loaded solid lipid nanoparticles for antifungal infection. The structure, morphology and size of resultant SCZ-SLNs were characterized by TEM, DSC, XRD, FTIR and so on. The antifungal activity was also tested in vitro and in vivo approach. The work is interesting and valuable. Reviewer recommends publication of this paper in this journal after a major revision. Specific concerns are attached as following:

(1) From Figure 2, the size of SCZ-SLNs was not uniform;

(2) Please keep Y-axis of Figure 4A-D consistent;

(3) The discussion of X-ray diffraction spectra is not correct. X-axis is angle.

(4) Section 3.2.3 is the same as Section 3.2.4.

Author Response

(1) From Figure 2, the size of SCZ-SLNs was not uniform

Size range is defined in the Manuscript and particle size obtained are in the range as described

(2) Please keep Y-axis of Figure 4A-D consistent

Thank you for your suggestion. The Y-axis of Figure 4A-D has been standardized to ensure consistency.

(3) The discussion of X-ray diffraction spectra is not correct. X-axis is an angle.

Thank you for your valuable comment. The discussion has been corrected.

(4) Section 3.2.3 is the same as Section 3.2.4.

Thank you for pointing out. The following section is removed.

Reviewer 2 Report

Comments and Suggestions for Authors

The manuscript “Sulconazole loaded solid lipid nanoparticles for enhanced antifungal activity, In Vitro and In vivo approach.'' by Ayesha Samee1, et al. the paper outlines a study on the formulation of solid lipid nanoparticles (SLNs) with Sulconazole (SCZ) as a model hydrophobic drug for the treatment of fungal infections. The very first thing I want to say is that this is a very good and necessary manuscript. And this article is well written, logical, well-illustrated and contains many interesting facts.  The topic of research is relevant and I recommended acceptance after addressing some minor comments. Here are some comments and suggestions for the authors to consider:

1.     Abstract, Line 30, zeta potential (-36.98) , this value not mentioned in the result part , revise if the value correct ?

2.     Abstract , line 31-32, revise the sentence, as DSC,FTIR,XRD results not demonstrated the morphology of the particles.

3.     Section 2.6, line 229, the pharmacokinetic models. Rewrite the term it should be kinetic release model.

4.     Section 2.9, line 269, number of rabbits is eight or nine ?

5.     Section 2.10. line 282, mention from where you take the histopathology part, which part of rabbits?

6.     Section 2.12, you mention (oral toxicity data) and their no study in the paper for this study?

7.     Result section, for particle size and zeta potential of the optimized formula is better if you could provide figures for that .

8.     Section 3.1.5, XRD, the author mention the position of peaks at their corresponding intensity, it better to mention the peaks position with respect to their position on x-axis (2θ Degree) and also mention the height of the intensity .

9.     Section 3.2.4. it repeated, delete it.

10.  Line 479-480. Provide reference.

11.  Figure 10. Provide histopathology figures without cropping  .

12.  References.    The style should in journal style format.

Author Response

  1. Abstract, Line 30, zeta potential (-36.98) , this value not mentioned in the result part , revise if the value correct ?

Thank you for pointing out. The mentioned value is corrected.

  1. Abstract, line 31-32, revise the sentence, as DSC, FTIR,XRD results do not demonstrate the morphology of the particles.

           Thank you for highlighting. The sentence has been revised.

  1. Section 2.6, line 229, the pharmacokinetic models. Rewrite the term it should be kinetic release model

              Thank you for suggestion following sentence is revised.

  1. Section 2.9, line 269, number of rabbits is eight or nine?

 Thank you for highlighting. The number of rabbits were nine. The values are corrected in manuscript.

  1. Section 2.10. line 282, mention from where you take the histopathology part, which part of rabbits?

Thank you for highlighting. The rabbit’s back skin was used for histopathology study. Following section have been updated in manuscript.

  1. Section 2.12, you mention (oral toxicity data) and their no study in the paper for this study?

Thank you for pointing out. Oral toxicity study was not performed. We have revised section 2.12.

  1. Result section, for particle size and zeta potential of the optimized formula is better if you could provide figures for that

Figure is added as suggested by Reviewers.

  1. Section 3.1.5, XRD, the author mention the position of peaks at their corresponding intensity, it better to mention the peaks position with respect to their position on x-axis (2θ Degree) and also mention the height of the intensity .

Thank you for your valuable comment. The peak position has been incorporated and the section       has been corrected.

  1. Section 3.2.4. it repeated, delete it.

Thank you for pointing out. The following section is removed.

  1. Line 479-480. Provide reference

Reference added as suggested

  1. Figure 10. Provide histopathology figures without cropping 

Thank you for suggestion, Figures are revised in manuscript.

  1. References.    The style should in journal style format.

            Thank you for your suggestion. References are updated in manuscript.

Reviewer 3 Report

Comments and Suggestions for Authors

Minor errors (correct or rewrite):

25 and 595. We report synthesis and optimization of SLNs. Synthesize is not an appropriate word

35 and 255 and 267 carbapol based

66. Connotations cannot

73. mechanism result in recurrent recurrence

133 reflective index

395. Figure 5. SLN4 (A), GMS (B), Phospholipon® 90H (C) and D (Sulconazole). SLN2 or SLN4???

405 the share rate and

Major corrections

93. The cross-networking of the gel prevents the polymorphic transition and aggregation of SLNs. Which polymorphic transitions?????

194. 2.5.2. Rheological studies . Method decription not compatible with results shown in fig 6. 50 rpm are referred only. If rpm didn´t change how did you get different shear rates? Which spindle was used?

317 3.1.2. FTIR Spectroscopy. Conclusions “FTIR analysis indicated chemical compatibility of SCZ, Phospholipon® 90H and GMS inside SLN2 formulation.” Can not be obtained by this analysis or spectra. What exactly evidences chemical compatibility???

346. 3.1.4. Thermal analysis. Interpretation of the mixture DSC thermogram is not correct, nor in agreement with the previous one. I see an additive effect which may allow to conclude that the three compounds are chemically compatible. However that is not explained.

The same applies to the mixture TG thermogram. Again I see an additive behaviour indicating chemical compatibility which is not explained properly.

Comments on the Quality of English Language

NA

Author Response

Minor errors (correct or rewrite):

25 and 595. We report synthesis and optimization of SLNs. Synthesize is not an appropriate word

Thank you for your suggestion. The following sentence has been revised.

35 and 255 and 267 carbapol based

Thank you for your suggestion. The spelling of Carbopol was corrected in the whole manuscript.

  1. Connotationscannot

 Thank you for your suggestion. The following sentence has been revised.

  1. mechanism result in recurrent recurrence

133 reflective index

Thank you for your suggestion. The following sentence has been revised.

  1. Figure 5SLN4(A), GMS (B), Phospholipon® 90H (C) and D (Sulconazole). SLN2 or SLN4???\

Thank you for pointing out. It was SLN2. The formulation code has been revised in manuscript.

405 the share rate and

 Thank you for your suggestion. The following sentence has been revised.

Major corrections

  1. The cross-networking of the gel prevents the polymorphic transitionand aggregation of SLNs. Which polymorphic transitions?????

The polymorphic transition referred to the change in the shape of the nanoparticles from spherical (a-form) to non-spherical (b-form) when they undergo the polymorphic transition. This polymorphic transition can lead to the aggregation and gelation of solid lipid nanoparticle (SLN) suspensions. The cross-networking of the gel prevents this polymorphic transition and aggregation of SLNs. Polymorphic transitions in lipid nanoparticles can lead to alterations in lipid packing and the internal structure of the nanoparticles, which may have negative consequences for drug loading. However, the use of solid lipid nanoparticles incorporated in gel form has been shown to enhance the skin accumulation and uptake of antifungal agents.

References:

Helgason, T., Awad, T. S., Kristbergsson, K., McClements, D. J., & Weiß, J. (2008). Influence of polymorphic transformations on gelation of tripalmitin solid lipid nanoparticle suspensions. Journal of the American Oil Chemists' Society, 85(6), 501-511. https://doi.org/10.1007/s11746-008-1219-9

Bunjes, H. (2010). Lipid nanoparticles for the delivery of poorly water-soluble drugs. Journal of Pharmacy and Pharmacology, 62(11), 1637-1645. https://doi.org/10.1111/j.2042-7158.2010.01024.

  1. 2.5.2. Rheological studies. Method decription not compatible with results shown in fig 6. 50 rpm are referred only. If rpm didn´t change, how did you get different shear rates? Which spindle was used?

Thank you for highlighting. Rheology study was performed by employing spindle number 64, applying varying torque values to the gel. The measurements for each sample were taken across a range of shear rates from 2 to 100 s−1. We have revised the sentence.

317 3.1.2. FTIR Spectroscopy. Conclusions “FTIR analysis indicated chemical compatibility of SCZ, Phospholipon® 90H and GMS inside SLN2 formulation.” Can not be obtained by this analysis or spectra. What exactly evidences chemical compatibility???

Compatibility is suggestive of possible interaction between drug and formulation components. We have corrected in manuscript as chemical compatibility with compatibility for better understanding and clarity.

  1. 3.1.4. Thermal analysis. The interpretation of the mixture DSC thermogram is not correct, nor in agreement with the previous one. I see an additive effect which may allow us to conclude that the three compounds are chemically compatible. However, that is not explained.

The same applies to the mixture TG thermogram. Again, I see an additive behavior indicating chemical compatibility which is not explained properly.

Thank you for the comment. The interpretation has been incorporated accordingly and explained properly.

Round 2

Reviewer 3 Report

Comments and Suggestions for Authors

Author have properly addressed all comments and corrected what needed accordingly